# The Effect of Farming Management and Crop Rotation Systems on Chlorophyll Content, Dry Matter Translocation, and Grain Quantity and Quality of Wheat (*Triticum aestivum* L.) Grown in a Semi-Arid Region of Iran

Aram Gorooei [1,2], Thomas Gaiser [1,*], Amir Aynehband [2], Afrasyab Rahnama [2] and Bahareh Kamali [1]

[1]  Institute for Crop Science and Resource Conservation (INRES), University of Bonn, Katzenburgweg 5, 53115 Bonn, Germany

[2]  Plant Production and Genetics Department, Faculty of Agriculture, Shahid Chamran University of Ahvaz, Ahvaz 6135783151, Iran

*  Correspondence: tgaiser@uni-bonn.de; Tel.: +49-228-73-2050

**Abstract:** To find suitable farming management approaches in the semi-arid climate of Iran, we set up an experiment combining three farm management practices with four crop rotation systems over four growing seasons (two winter and two summer seasons), from 2018 to 2020. The three farm management practices comprised: intensive (IF, with inorganic inputs, removal of crop residues from the soil, and weeds chemically controlled), organic (OF, with organic inputs, a return 30% of crop residues in the soil, and weeds mechanically controlled), and integrated (INT, with mineral/organic inputs, return 15% of crop residues to the soil, integrated weed control). The four crop rotation systems were: fallow-wheat (F-W), maize-wheat (M-W), sesame-wheat (S-W), and mung bean-wheat (B-W). Treatment effects were assessed by chlorophyll (Chl) content, photosynthetic parameters, and wheat grain quality and quantity measurements. All management practices from the first to the second year resulted in increases in the total *Chl* content and post-anthesis photosynthesis (PAP). The total *Chl* content under INT was improved through a greater increase in *Chl-b* compared to *Chl-a*. Dry matter remobilisation (DMR) was higher under INT than under IF. The highest (39) and lowest (23) grain number per spike were obtained in IF under B-W and OF under F-W, respectively. B-W produced the highest grain yield (541.4 g m$^{-2}$). The protein contents in farming with organic matter inputs were higher than that under IF. INT produced an optimum level of wheat yield despite a 50% reduction in chemical inputs, and this was achieved through the fast absorption of chemical elements at the beginning of growth, and having access—at the grain filling stage—to elements derived from organic matter decomposition, and through the utilisation of DMR. Our results indicate that implementing B-W and S-W under INT is a promising strategy for this region. However, the results need to be further evaluated by long-term experiments.

**Keywords:** integrated farming management; crop residue; chlorophyll; wheat production; remobilisation; protein content; crop rotation system; organic matter



## 1. Introduction

Increasing crop yields of recent decades have been strongly linked to the intensive use of chemical inputs, e.g., fertilisers, herbicides, and pesticides [1]. Such intensive farming practices have been implemented due to population growth, and the resulting pressure on agricultural lands has changed the soil nutrient status [2], soil and plant biodiversity [3], and agricultural landscapes [4]. Intensive farming management (IF) involves the intensive use of chemical fertilisers [5], removal of crop residues [6], and deep tillage to prepare the seed bed [7], all of which lead to high productivity [8]. In contrast to IF, the set of management practices implemented in organic farming (OF), e.g., returning crop residues

to the soil, using organic and biological fertilisers, controlling weeds bio-mechanically, conservation tillage, applying different crop rotation systems, and intercropping [9,10], lead to improved recycling of soil nutrients, enhanced soil and crop biodiversity [11], and better yield quality [12]. However, the average productivity of OF is lower than that of IF, and it may not be sufficient for the future food needs of a growing population. Integrated farming (INT), which has been offered as an intermediate solution, is the integration of IF and OF management practices in a manner that focuses on high quantity and quality of crop production, environmentally friendly cultivation [13], and long-term maintenance of soil health and fertility to ensure food security [14].

Many studies have investigated ways to enhance crop productivity through the appropriate use of mineral and organic fertilisers, crop residue management, selection of optimal crop species adapted to specific conditions, and the replacement of pesticides with bio-mechanical methods of pest control [15–17]. However, the use of organic matter (e.g., compost and crop residues) in the soil reduces grain yield due to the immobilisation of nutrients, especially during the initial growth stage of plants. This situation can be improved by the combined use of these organic resources with inorganic fertilisers, which can improve wheat (*Triticum aestivum* L.) grain yield by positively affecting the biological–chemical–physical properties of the soil [18]. For example, the application of organic and inorganic nutrient resources to wheat improved the nitrogen supply and remarkably enhanced wheat yield components (e.g., 1000-grain weight), tillers, wheat yield, and nitrogen uptake by the grain compared to wheat supplied with only inorganic fertilisers [19,20].

Apart from farming management, designing a proper crop rotation leads to sustainable crop yields [21]. Some of the positive effects of implementing crop rotation systems are: improved and stable crop yield, high soil nutrient availability, increased nutrient use efficiency, improved soil biochemical and structural properties, as well as reductions in weed stress and pest invasion incidence [22,23]. Geng et al. [1] reported that the yields of both wheat and maize (*Zea mays* L.) under long-term crop rotation systems were significantly higher than those of each monocultured crop. Similarly, double cropping systems based on winter wheat and summer crops, e.g., sorghum (*Sorghum bicolor* L.) and sesame (*Sesamum indicum* L.) improved annual net production and long-term sustainability compared to a summer-fallow rotation system [24]. In addition, soybean (*Glycine max* L.) grown in rotation with wheat was more successful in reducing the incidence of pests and increasing wheat productivity compared to a succession of maize-wheat [23]. Furthermore, a comparison of different cropping systems comprising wheat, soybean, and maize to monocropped wheat revealed that wheat grain yield obtained from monocropping was remarkably lower than those of the other cropping systems [25]. The results of similar studies show that including legumes in cereal cropping systems (e.g., wheat) barely (*Hordeum vulgare* L.) improves the grain yield of wheat [26,27].

Farming management and crop rotation affect crop productivity by influencing factors such as soil properties and plant characteristics such as leaf chlorophyll (Chl) content and its photosynthetic activity [1]. The production of dry matter through the photosynthetic activity of leaves is the most important source of grain filling in cereals. *Chl* content, which reflects the health of crop leaves, plays a crucial role in the photosynthetic process and therefore dry matter production [28,29]. The slow but steady release of minerals from organic matter resources increases cereal grain yield by improving the photosynthetic capacity and *Chl* content, especially during flowering and grain filling [1,30]. Iqbal et al. [29] noted that a combination of organic manure and chemical fertiliser application increases the levels of *Chl-a* and *Chl-b* and also increases the production of photoassimilates. Aside from photosynthesis, crop yields can also be improved by remobilising dry matter accumulated in vegetative parts before anthesis to the grains during grain filling [31,32]. However, *Chl* content and the translocation of photoassimilates within a plant can be affected by a variety of farming practices and climate conditions [30,33,34]. Therefore, investigating dry matter production and its translocation, particularly during grain filling, is important to optimise farming management, e.g., timing and amount of fertiliser application.

Different agricultural practices are known to impact wheat grain yield [15,21,35]; however, little is known regarding how *Chl* content, carbon assimilation, and remobilisation of pre-anthesis dry matter contribute to wheat grain yield under different farm management practices in combination with various wheat-based rotation systems in the semi-arid regions of Iran.

Because of this knowledge gap, we aimed to (1) determine the effect of different combinations of organic matter (compost, vermicompost, crop residue, humic acid) and inorganic fertilisers (N-P-K) on *Chl* content and photosynthetic parameters; (2) investigate how the quality and quantity of wheat grains respond to different double cropping systems of winter wheat and summer crops; and (3) study the effects of various combinations of fertiliser/crop residue management practices and crop rotation systems on post-anthesis biomass re-translocation and wheat grain yield.

## 2. Materials and Methods

### 2.1. Study Area

The field experiment was conducted at the field research farm of Shahid Chamran University, Ahvaz, Khuzestan province (48°41 E and 31°20 N, altitude: 22.5 m), located in the southwestern part of Iran (Figure 1a). Khuzestan province produces 12% of the country's wheat, and it is known for the high quantity and quality of its wheat grain production [36]. The average rainfall in the region is 213 mm yr$^{-1}$, 85% of which falls during the winter wheat growing season. The average temperature is 25 °C, with the maximum and minimum temperatures of 48 °C and 4 °C occurring in the summer and winter, respectively [37] (Figure 1b). The dominant soil texture in the topsoil at the experimental site is sandy-loam. Other topsoil properties are: total nitrogen content of 0.039%; total organic carbon of 0.452%; total phosphorus of 13 mg kg$^{-1}$; potassium of 159 mg kg$^{-1}$; electrical conductivity of 3.4 dS m$^{-1}$; and pH of 7.8. This site was already under wheat cultivation and managed using both organic and inorganic fertilisers for three years before the start of the experiment.

### 2.2. Experimental Design

The experiment includes two factors, i.e., main factor of farming managements and subfactor of crop rotation systems (12 treatments). The main factors consisted of three combinations of fertiliser and crop residue management practices. The following farm management practices were included: (1) intensive farming (IF), in which 100% of inorganic inputs were used, and crop residues were completely removed from the soil after harvesting each crop and before planting the next crop; (2) organic farming (OF), in which 100% of organic and biological inputs were used, and 30 percent of crop residues from the preceding year were returned to the soil; (3) integrated farming (INT), in which a combination of inorganic, organic, and biological fertilisers was used, and only 15% of the crop residues from the preceding year were returned to the soil. More details about the three different farm management practices are given in Table 1.

Subfactors consisted of four types of crop rotation systems and comprised: fallow-wheat (F-W), maize-wheat (M-W), sesame-wheat (S-W), and mung bean (*Vigna radiata* L.) -wheat (B-W) (Figure 1c). The size of each experimental plot was 12 m$^2$ (3 m × 4 m) with three replications.

Under OF and INT, air-dried manure was evenly spread on the surface of the experimental plots and then mixed with the soil before planting the crops. Table S1 presents the detailed chemical properties of the two types of manure used in the experiment as well as the crop residues returned to the soil. The organic and biological fertilisers used in OF were also used in INT. The experiment began in July 2018 with the cultivation of summer crops and ended in May 2020 (Figure 1c). The furrow planting method was used to cultivate summer crops. After harvesting the summer crops, wheat was planted in rotation in rows with inter- and intra-row spacing of 7 and 2 cm, respectively. At the beginning of the experiment, a mouldboard plough was used to prepare furrows. Subsequently, to

prevent mixing between the experimental plots, a shovel and rake were used to prepare furrows for summer crops and rows for wheat. Complete descriptions of all agronomic operations, such as planting date, the types of organic and inorganic fertilizers, plant density, harvest date, cultivar type, as well as the method of cultivation, are provided in our previous work "[36]". Additionally, descriptions of the implementation of these operations are summarised in the supplementary information (Tables S1 and S2).

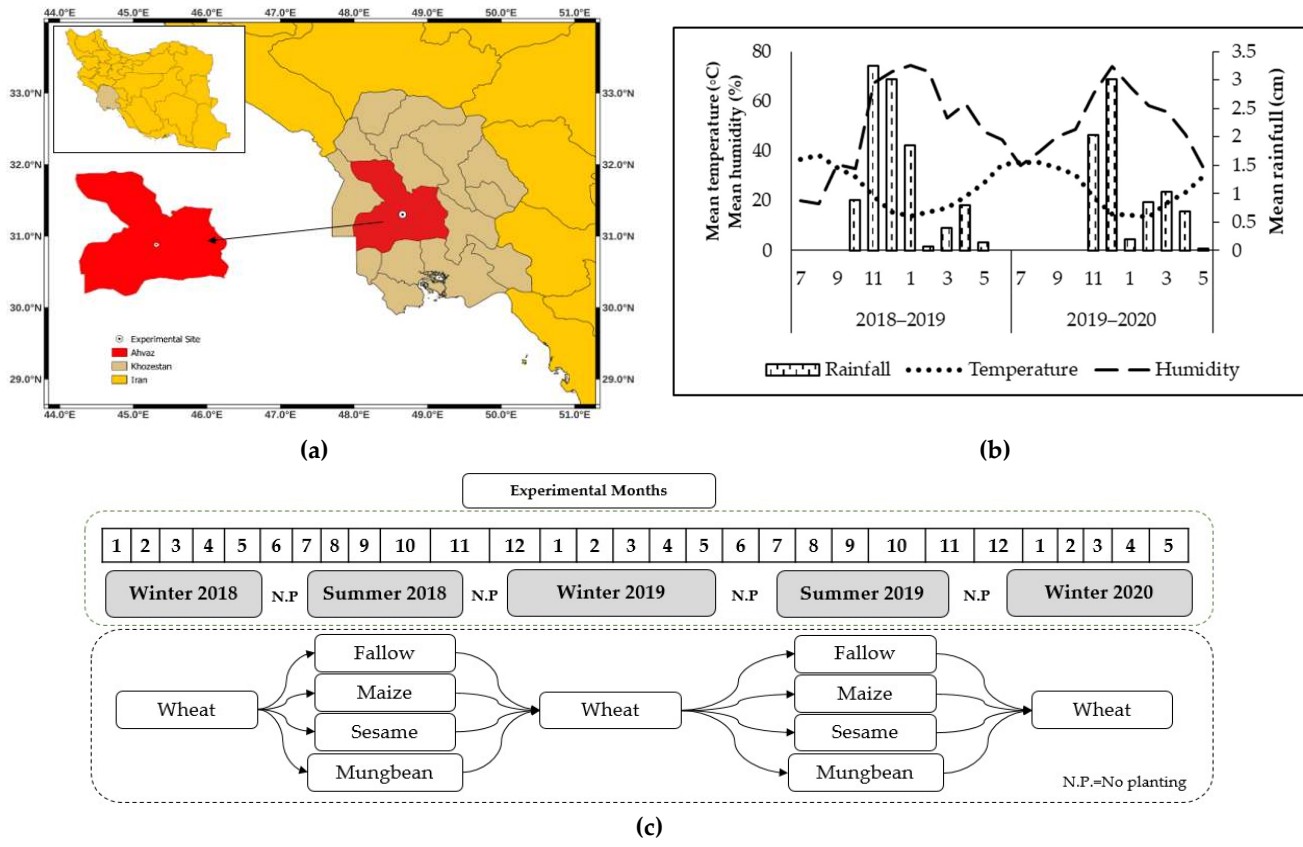

(a)

(b)

(c)

**Figure 1.** (**a**) Location of Khuzestan province and the experimental site in Iran, (**b**) average monthly rainfall, average temperature, and humidity in the experiment site during the period of 2018–2020, and (**c**) schematic diagram of the cultivation calendar for winter and summer crop rotation systems during the study period. N.P. = no planting.

**Table 1.** Complete information on applied organic/biological (air-dried vermicompost and compost, humic acid, and biological phosphate) and inorganic fertilisers (N-P-K), crop residues, herbicides, and fungicides in the three farm management practices consisting of intensive (IF), integrated (INT), and organic (OF) practices. The four crops consisted of wheat (W), mung bean (B), maize (M), and sesame (S). The experiment lasted from 2018 to 2020 at the experimental plots in Ahvaz, Khuzestan province, Iran.

| | IF | | | | OF | | | | INT | | | |
|---|---|---|---|---|---|---|---|---|---|---|---|---|
| | **W** | **B** | **M** | **S** | **W** | **B** | **M** | **S** | **W** | **B** | **M** | **S** |
| Organic/biological fertilisers | | | | | | | | | | | | |
| Vermicompost (t ha$^{-1}$) | - | - | - | - | 3.3 | 3.3 | 3.3 | 3.3 | 1.7 | 1.7 | 1.7 | 1.7 |
| Manure compost (t ha$^{-1}$) | - | - | - | - | 10 | 10 | 13.3 | 10 | 5 | 5 | 6.7 | 5 |
| Humic acid (kg ha$^{-1}$) [1] | - | - | - | - | 20 | - | - | - | 10 | - | - | - |
| Biological phosphate (l ha$^{-1}$) [2] | - | - | - | - | 4 | - | - | - | 2 | - | - | - |

**Table 1.** *Cont.*

| | IF | | | | OF | | | | INT | | | |
|---|---|---|---|---|---|---|---|---|---|---|---|---|
| | **W** | **B** | **M** | **S** | **W** | **B** | **M** | **S** | **W** | **B** | **M** | **S** |
| Crop residues return to soil (t ha$^{-1}$) | Removed from the soil | | | | 0.9 | 2.88 | 3.93 | 1.29 | 0.45 | 1.44 | 1.965 | 0.645 |
| Inorganic fertilisers | | | | | | | | | | | | |
| Nitrogen (kg ha$^{-1}$) [3] | 110 | 30 | 200 | 75 | - | - | - | - | 55 | 15 | 100 | 37.5 |
| Phosphorus (kg ha$^{-1}$) | 100 | 50 | 100 | 50 | - | - | - | - | 50 | 25 | 50 | 25 |
| Potassium (kg ha$^{-1}$) | 100 | 50 | 100 | 50 | - | - | - | - | 50 | 25 | 50 | 25 |
| Control of weed [4] | 10 L ha$^{-1}$ Roundup | - | - | - | Manually | | | | 5 L ha$^{-1}$ Roundup+Manually | Manually | | |
| Fungicide (g kg$^{-1}$ seed) [5] | 160 | | 45 | | - | - | - | - | 80 | - | 23 | - |

[1] Humic acid was sprayed at the wheat flowering stage at a concentration of 5 mL L $^{-1}$. [2] One kg of seeds was soaked in 25 mL biological phosphate (3R-BioPhosphate: *Pseudomonas putida* and *Pantoea agglomerans*: $10^7$ bacteria g$^{-1}$) one hour before planting. [3] Two-thirds of nitrogen fertiliser was applied at planting time and one-third at the tillering stage of wheat growth. [4] The dominant weed was *Cynodon dactylon* L. [5] Difenoconazole was sprayed on the seeds before planting.

### 2.3. Sampling and Analysis

#### 2.3.1. Determination of Chlorophyll Content and Photosynthesis Parameters

The contents of *Chl-a*, *Chl-b*, and total *Chl* were measured from leaf samples collected from the spot closest to the wheat spike at the flowering stage. Twenty leaves were randomly selected from each experimental plot. All samples of fresh leaves were cut into small pieces, and one gram of freshly chopped leaves was placed in a volumetric flask containing 10 mL of 80% acetone solution and kept in the dark for 24 h [38,39]. The absorbance of the extracted solution was measured with spectrophotometer (S2100, UV-Vis, UNICO, Fairfield, NJ, USA) at 663 and 645 nm. The contents of *Chl-a*, *Chl-b*, and total *Chl* were calculated using the following equations [40,41]:

$$Chl - a \left[ \frac{mg}{g\ fresh\ leaf} \right] = [(12.7 \times A_{663}) - (2.69 \times A_{645})] \times \left( \frac{V}{1000} \times W \right) \tag{1}$$

$$Chl - b \left[ \frac{mg}{g\ fresh\ leaf} \right] = [(22.9 \times A_{645}) - (4.68 \times A_{663})] \times \left( \frac{V}{1000} \times W \right) \tag{2}$$

$$Total\ Chl \left[ \frac{mg}{g\ fresh\ leaf} \right] = Chl - a + Chl - b \tag{3}$$

where *A* is the absorbance of the sample solution at the aforementioned wavelengths.

To determine how much of the dry matter in wheat grains originated from either photosynthesis during grain filling or from the remobilisation of assimilates in vegetative organs (stems and leaves) during grain filling, the vegetative parts were weighed at anthesis and maturity. The date of anthesis was defined as the time when 50% of the spikelets within a spike showed visible anthers [42]. At both growth stages, 20 plants were randomly cut from the centre of each plot at the ground level. The vegetative parts, including stems and sheaths, leaves, and glumes (spike axis and kernel husks) were separated. All samples were then dried at 75 °C until they reached a constant weight [43]. The dry weights of vegetative parts at anthesis and maturity were used to calculate dry matter remobilisation during grain filling (DMR: re-transfer of dry matter stored in vegetative parts after anthesis to the grains) and post-anthesis-produced carbohydrates (PAP: grain filling carbohydrates that originate from photosynthesis) [32,42,44] using the following equations:

$$\mathrm{DMR} \left[ \frac{g}{plant} \right] = \mathrm{DM_{Anthesis}} - [\mathrm{DM}_{leaves+\ culms\ +\ chaff}]_{maturity} \tag{4}$$

$$\text{PAP}\left[\frac{g}{plant}\right] = [\text{Grain Yield}] - [\text{DMR}] \tag{5}$$

### 2.3.2. Measurement of Quantity and Quality of Grain Wheat

In addition to grain yield, we measured three yield components: spikes per m$^2$, the number of grains per spike, and 1000-grain weight. Measurements were conducted on plants that were collected after maturity from the centre of each experimental plot measuring 4 m$^2$. The samples were then dried in the oven at 75 °C for 48 h. Grain quality was assessed by measuring the concentration of nitrogen in the grains according to the Kjeldahl method. Protein content was calculated from the nitrogen concentration [45] using the following equation:

$$\text{Grain protein }[\%] = \text{Grain nitrogen }[\%] \times 5.7 \tag{6}$$

where the value of 5.7 refers to the protein factor for wheat [46].

### 2.4. Statistical Analyses

The effects of time (year), crop rotation system, farming management and their interaction on *Chl* contents, dry matter translocation, and yield components were evaluated by analysis of variance (ANOVA) using SAS version 9.4, ref. [47] implementing the mixed model. The effect of year was considered to be random, and the effects of the crop rotation system and farming management were considered fixed. To compare and explain the results, the means of the interactions between two and three of the experimental factors were considered. Statistical mean comparisons between treatments were evaluated by the least significant difference at a probability (*p*) level of 0.05 and 0.01 by the Duncan method [48].

### 3. Results

### *3.1. Effect of Year and Treatment Interactions on Chlorophyll Contents*

The year, farming management, crop rotation system, as well as their interaction all had significant ($p \leq 0.01$) effects on *Chl-a*, *Chl-b*, and total *Chl* (Table 2). The *Chl-a* content increased significantly from the first experimental year to the second under the interaction effects of the three treatments. The highest concentration of *Chl-a* (1.32 mg g$^{-1}$ fresh leaf) was obtained in F-W, followed by S-W (1.22 mg g$^{-1}$ fresh leaf) and B-W (1.26 mg g$^{-1}$ fresh leaf) under IF during the second year of the experiment (Figure 2a). INT and OF had no statistically significant effect on *Chl-a* in all crop rotation systems in both years of the experiment, with the exception of S-W under OF, which produced a significant increase in *Chl-a* content. Similarly, levels of *Chl-b* were higher in IF than in OF (Figure 2a,b).

Interestingly, *Chl-b* levels under INT increased significantly in the second year of the experiment; moreover, under INT and most of the crop rotation systems, *Chl-b* levels did not differ significantly ($p \leq 0.05$) from those of IF (Figure 2b). IF management with B-W and M-W crop rotation systems produced the highest amounts of total Chl, with values of 1.94 and 1.8 mg g$^{-1}$ fresh leaf, respectively (Figure 2c). Meanwhile, the lowest content of total *Chl* was obtained under OF management with F-W crop rotation (1.3 mg g$^{-1}$ fresh leaf).

**Table 2.** Effects of year (Y), farm management practice (F), crop rotation system (CS), and their interactions on chlorophyll-a, chlorophyll-b, total chlorophyll, dry matter remobilisation (DMR), and post-anthesis dry matter photosynthesis (PAP) in winter wheat during the period of 2018–2020, as determined by ANOVA.

| Source | df | *Chl-a* | *Chl-b* | Total Chl | DMR | PAP |
|---|---|---|---|---|---|---|
| Year (Y) | 1 | 0.51 ** | 0.35 ** | 1.7 ** | 1726 NS | 62078 ** |
| Farming management practice (F) | 2 | 0.13 ** | 0.191 ** | 0.53 ** | 6753 ** | 152919 ** |
| Crop rotation system (CS) | 3 | 0.012 * | 0.045 ** | 0.05 * | 9044 ** | 10988 ** |
| Y × CS | 3 | 0.041 * | 0.03 ** | 0.04 NS | 1509 NS | 1172 NS |
| Y × F | 2 | 0.054 * | 0.007 NS | 0.02 NS | 1918 * | 1727 NS |
| F × CS | 6 | 0.023 NS | 0.015 ** | 0.03 NS | 2622 ** | 6588 ** |
| Y × F × CS | 6 | 0.053 ** | 0.012 ** | 0.09 ** | 1749 * | 2584 * |
| Error Y (Y × R) | 4 | 0.023 | 0.002 | 0.03 | 188 | 688 |
| Error F × Y (F × Y × R) | 8 | 0.0121 | 0.0013 | 0.013 | 991 | 805 |
| Error | 36 | 0.012 | 0.002 | 0.017 | 532 | 970 |
| C.V | | 11.16 | 12.1 | 9.2 | 15.5 | 9.08 |

*, **, and NS display the level of significant at $p \leq 0.05$, $p \leq 0.01$, and no significant difference by Duncan test, respectively.

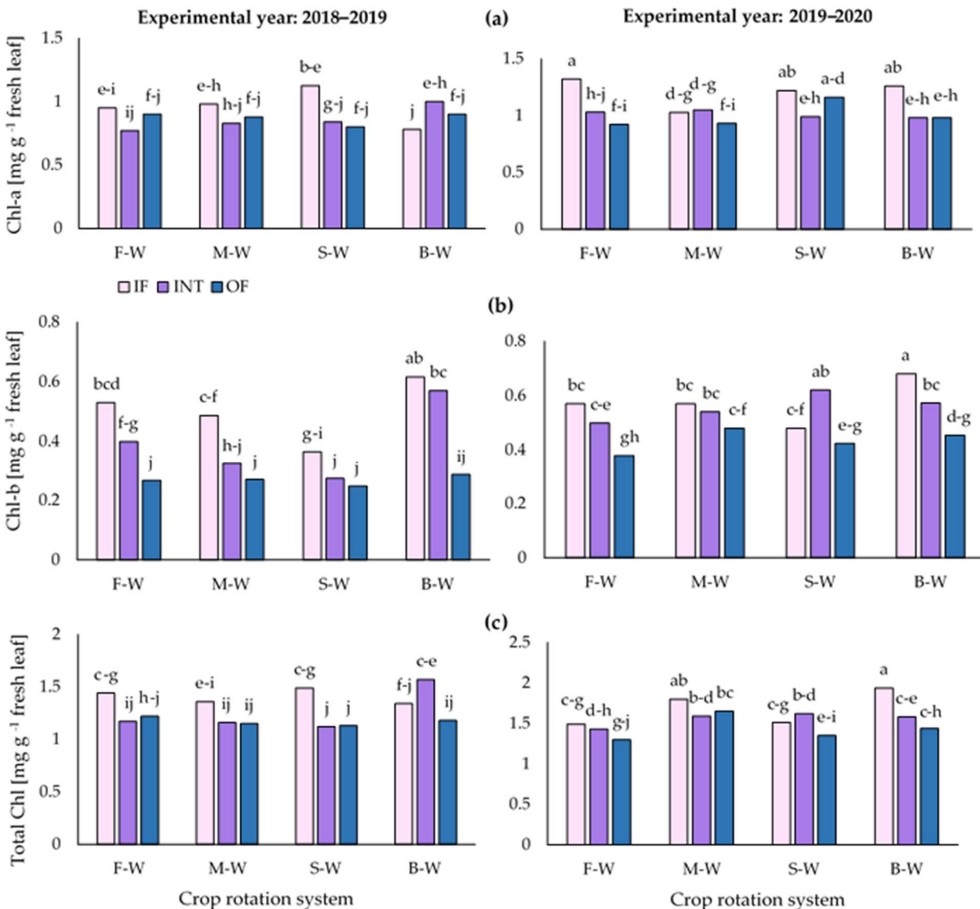

**Figure 2.** Effects of the interaction between three farm management practices (intensive, IF; organic, OF; and integrated, INT) and four crop rotation systems (fallow-wheat, F-W; maize-wheat, M-W; sesame-wheat, S-W; and mung bean-wheat, B-W) on (**a**) chlorophyll-a, (**b**) chlorophyll-b and (**c**) total chlorophyll at flowering time during the two-year experiment. Means were compared by Duncan's test, and different letters above each bar indicate a significant difference at $p \leq 0.05$.

### 3.2. Effects of Year and Treatment Interactions on Dry Matter Translocation Parameters

All treatments and their interactions (i.e., year × farming management × crop rotation system) had significant ($p \leq 0.01$ and $p \leq 0.05$) effects on DMR and PAP, except for the individual effect of year, which had no effect on DMR (Table 2). DMR played a positive role in grain filling under the INT strategy implementing any of the crop rotation systems, but particularly M-W (Figure 3a). M-W under OF and F-W under IF produced the lowest DMR values of 127 and 95 mg plant$^{-1}$, respectively (Figure 3a). IF management had a more pronounced role in the production and translocation of dry matter from photosynthesis (PAP in grain filling, Figure 3b). In general, the three farm management practices produced PAP in the following descending order: IF > INT > OF. The combination of B-W and IF produced the highest PAP (Figure 3b). Overall, the differences described above can be seen by comparing the means of the individual effects of the experimental treatments (Figures S1a,b and S2a,b).

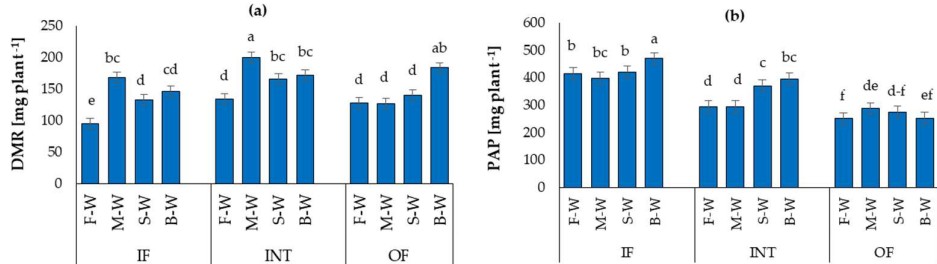

**Figure 3.** Effects of interaction between three farm management practices (intensive, IF; organic, OF; and integrated, INT) and four crop rotation systems (F-W, fallow-wheat; M-W, maize-wheat; S-W, sesame-wheat, and B-W, mung bean-wheat) on (**a**) DMR (dry matter remobilisation), and (**b**) PAP (post-anthesis dry matter photosynthesis). Means were compared by Duncan's test, and different letters above each bar indicate a significant difference at $p \leq 0.05$.

In contrast to IF, the interaction between year, farming management, and CS under INT and OF management and most of the CSs resulted in increased DMR and PAP values in the second year of the experiment (Figure 4a). The strong role of INT on DMR improvement was more apparent under S-W. Additionally, the lowest values of DMR were observed under IF implementing F-W. The PAP values further increased from 230, 394, and 317 mg plant$^{-1}$ with the ratio of 1.3, 1.18, and 1.11 in the second year under OF, IF, and INT, respectively (Figure 4b). PAP increased in all CSs in the second year of the experiment (Figure 4b), although we observed no differences between M-W and B-W under IF and B-W under INT. In the 2019–2020 cropping seasons, the lowest PAP was obtained in B-W under OF, with a value of 272 mg plant$^{-1}$ (Figure 4b).

### 3.3. Effect of Year and Interaction of the Treatments on Wheat Yield Components

ANOVA results reveal that the effect of each factor, i.e., year, farming management, and CS on wheat grain yield, grain number per spike, and 1000-grain weight was statistically significant at both $p \leq 0.01$ and $p \leq 0.05$ (Table 3). In contrast, these treatments did not affect the number of spikes per square metre significantly (Table 3). The interaction effect of the three treatments was not significant for all target variables, and therefore, the mean effect of farming management × CS on the quality and quantity of grain wheat, which was significant for most of the measured variables (Table 3), was also considered. Comparing the means of farming management × CS reveals the highest (39) and lowest (33) grain numbers were obtained under IF implementing B-W and F-W, respectively. Under IF management, we observed no significant difference in the grain numbers of M-W and S-W (Figure 5a). Under INT management, the S-W grain number was higher than that of M-W, but it was not significantly different from that of B-W (Figure 5a). The highest 1000-grain weight (40.6 g) was obtained under OF implementing S-W, while the lowest (36.79 g) was observed under INT implementing M-W (Figure 5b). Under INT and OF, the grain yields

for wheat cropping systems in rotation with mung bean and sesame were higher than those for maize CSs as well as that of monocropped wheat. Under IF, meanwhile, the grain yields were higher with B-W and M-W than with S-W and F-W (Figure 5c). In addition, the grain yields of S-W under IF and INT did not differ significantly ($p \leq 0.05$). An individual mean comparison of experimental treatments also provides the range of the difference in the target parameters under the influence of farming management and crop rotation systems (Figures S3a–d and S4a–d).

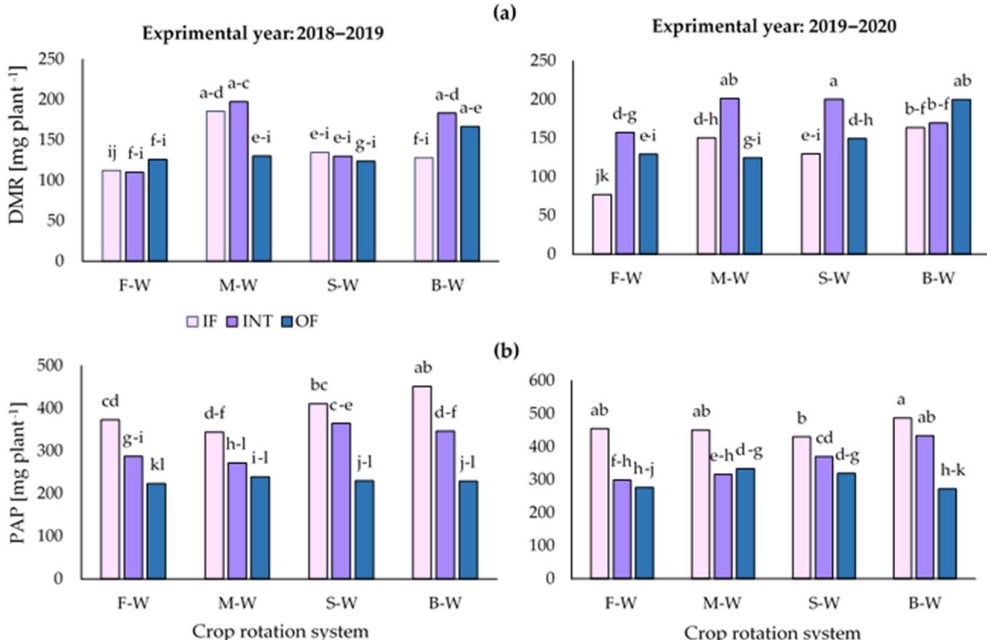

**Figure 4.** Effects of interactions between three farm management practices (intensive, IF; organic, OF; and integrated, INT) and four crop rotation systems (fallow-wheat, F-W; maize-wheat, M-W; sesame-wheat, S-W; and mung bean-wheat, B-W) on (**a**) dry matter remobilisation (DMR), and (**b**) post-anthesis dry matter photosynthesis (PAP) during the two-year experiment. Means were compared by Duncan's test, and different letters above each bar indicate a significant difference at $p \leq 0.05$.

**Table 3.** Effects of year (Y), farm management practice (F), crop rotation system (CS), and their interactions on the yield, yield components, and grain protein content in winter wheat during the period of 2018–2020, as determined by ANOVA.

| Source | df | Yield Components | | | | Quality Parameter |
| --- | --- | --- | --- | --- | --- | --- |
| | | Grain Number per Spike | 1000 Grain Weight | Spike per m$^{-2}$ | Grain Yield | Grain Protein |
| Year (Y) | 1 | 194.8 ** | 53.102 ** | 13.34 NS | 93071.1 ** | 4.67 ** |
| Farm management practice (F) | 2 | 692.9 ** | 9.54 ** | 7.59 NS | 148671 ** | 7.88 ** |
| Crop rotation system (CS) | 3 | 115.4 ** | 4.8 * | 42.9 NS | 33530 ** | 9.25 ** |
| Y × CS | 3 | 7.11 ** | 3.44 NS | 4.16 NS | 959 NS | 1.31 NS |
| Y × F | 2 | 0.56 NS | 7.4 * | 1.93 NS | 1386.4 * | 2.58 * |
| F × CS | 6 | 8.99 ** | 5.5 ** | 10.37 NS | 3316 ** | 9.72 ** |
| Y × F × CS | 6 | 8.69 ** | 2.85 NS | 6.48 NS | 1959 ** | 1.12 NS |
| Error Y (Y × R) | 4 | 1.054 | 0.548 | 3.97 | 115.15 | 0.48 |
| Error F × Y (F × Y × R) | 8 | 1.4 | 1.35 | 6.28 | 208 | 0.31 |
| Error | 36 | 1.38 | 1.46 | 5.42 | 428.45 | 0.49 |
| C.V | | 3.73 | 3.09 | 0.65 | 4.19 | 7.2 |

*, ** indicate significantly different at $p \leq 0.05$ and $p \leq 0.01$, respectively; NS, no significant difference by Duncan test based on a mixed split-plot statistical design.

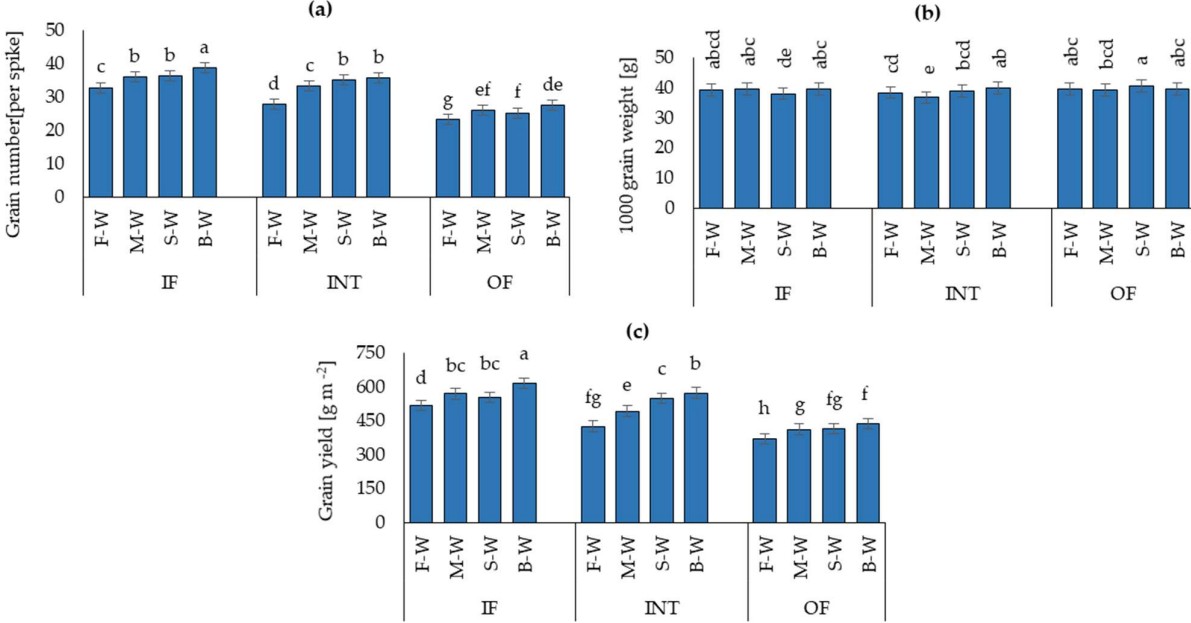

**Figure 5.** Interaction effect of three farm management practices (intensive, IF; organic, OF; and integrated, INT) and four crop rotation systems (fallow-wheat (F-W); maize-wheat (M-W); sesame-wheat (S-W), and mung bean-wheat (B-W) on (**a**) grain number per spike, (**b**) 1000-grain weight, and (**c**) grain yield. Different letters above each bar indicate a significant difference at $p \leq 0.05$.

Under the interaction effects of the three treatments, the values of all measured variables markedly increased in the second year of the experiment (2019–2020) from the previous year (2018–2019) (Figure 6a–c). However, the rates of increase varied depending on the yield component. The highest average number of grains per spike (41) was produced under IF management implementing B-W, while the lowest (22.2) was produced under OF implementing the S-W crop rotation system (Figure 6a). Meanwhile, we observed no significant difference ($p \leq 0.05$) between 1000-grain weights, with values varying between 37.5 and 42.9 g for most of the farming management and crop rotation systems (Figure 6b). Comparison of grain yields among all CSs showed the highest increases: under OF and INT implementing S-W, increasing by rates of 1.4 and 1.27, respectively; and under IF implementing M-W, which increased by a rate of 1.14 from the first to the second year (Figure 6c and Table S3).

### 3.4. Main and Interaction Effects of the Treatments on Wheat Grain Protein Content

The responses of grain protein to year, farming management, crop rotation system, and the interaction between farming management and CSs were significant at $p \leq 0.01$ (Table 3). Overall, the highest grain protein content was obtained for B-W with a value of 10.5%, followed by OF (10.35%), INT (9.56%), and IF (9.23%) (Figure S5a,b). Comparing means of the interaction effect of farming management and crop rotation system reveals the highest grain protein contents under INT implementing B-W and under OF implementing M-W and B-W (Figure 7, no significant differences between treatments). The lowest grain protein content (8.48%) was obtained under IF implementing M-W (Figure 7). Grain protein contents in M-W and S-W crop rotation systems under IF and INT were not significantly different ($p \leq 0.05$). The wheat grain protein contents of F-W and B-W were similar under IF management.

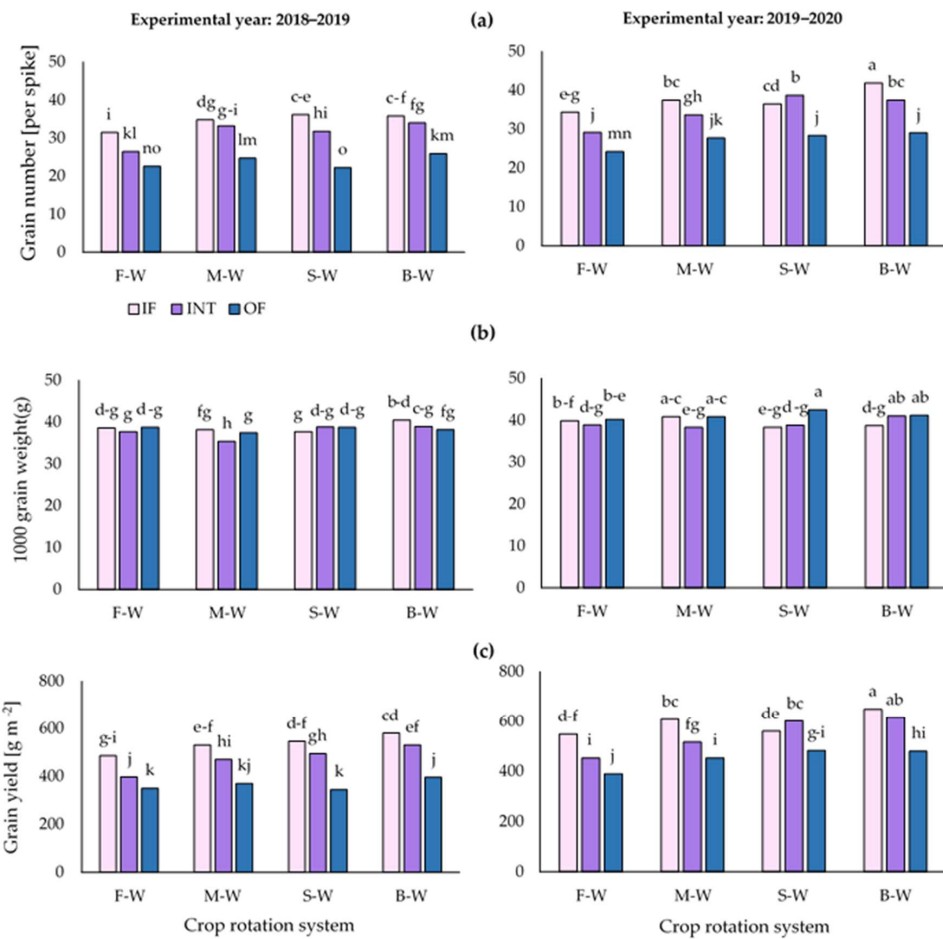

**Figure 6.** Effect of the interaction between three farm management practices (intensive, IF; organic, OF; and integrated, INT), and four crop rotation systems (fallow-wheat (F-W); maize-wheat (M-W); sesame-wheat (S-W) and mung bean-wheat (B-W) on (**a**) grain number per spike, (**b**) 1000-grain weight, and (**c**) grain yield during the two-year experiment. Different letters above each bar indicate a significant difference at $p \leq 0.05$.

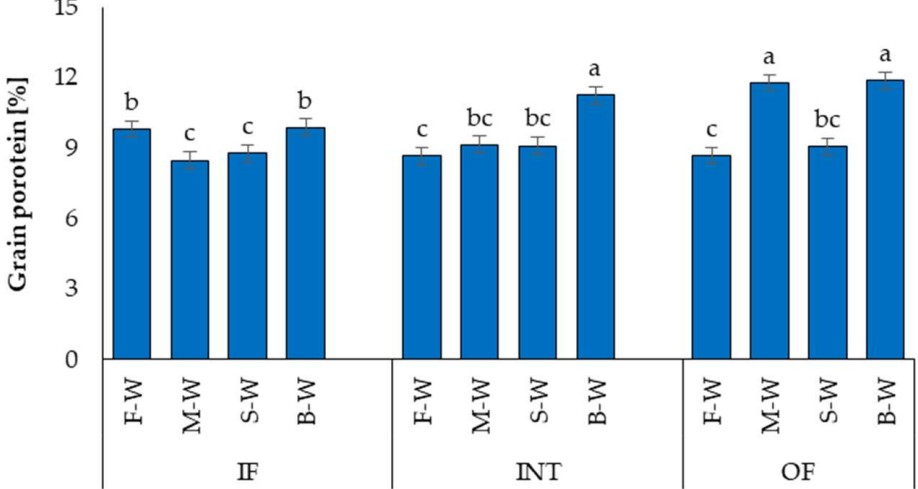

**Figure 7.** Effect of the interaction between three farm management practices (intensive (IF), organic (OF), and integrated (INT)) and four crop rotation systems (fallow-wheat (F-W), maize-wheat (M-W), sesame-wheat (S-W), and mung bean-wheat (B-W)) on average grain protein content over two years. Different letters above each bar indicate a significant difference at $p \leq 0.05$.

## 4. Discussion

*4.1. Effect of Year and Treatment Interactions on Chlorophyll Contents and Dry Matter Translocation*

*Chl* content is an index used to evaluate the health of leaves and the photosynthetic ability of crops, which is highly influenced by environmental conditions such as nutrition and temperature. Among the three farm management practices, the highest levels of *Chl-a* and total *Chl* were observed at the flowering stage under IF, which was due to higher nitrogen availability from mineral fertilisation, although we did not consistently observe this for all crop rotations and years. *Chl-b* content, on the other hand, showed a different trend: i.e., in the second year of the experiment, *Chl-b* levels under INT were as high as those under IF, and in fact, the former levels were even higher, except for crop rotation B-W. High contents of *Chl-b* maintain the continuity of photosynthetic activity, resulting in carbohydrate supplies going to physiological sinks (grains). Therefore, the difference in yields between IF and INT was smaller in the second year of the experiment, and in the crop rotation systems S-W and B-W, grain yields under INT were even higher than those under IF (Figure 6). These findings are consistent with the results reported by [49]. The increase in the content of Chls from the first experimental year to the next may be due to the gradual release of nutrients such as nitrogen and manganese from organic fertiliser sources (e.g., manure, biofertilisers, and humic acid) and their incorporation into loops of chlorophyll [50,51].

After the increase in *Chl* contents under IF, we observed an enhancement in photosynthetic sources (leaves, stems, sheaths), leading to higher PAP at the grain-filling stage [52–54]. The high leaf area index value under IF may be another reason that its PAP value was higher than those of the other farm management practices (Figure S3c). An extended wheat canopy favours light and carbon dioxide absorption, thus improving photosynthetic performance [55]. Favourable weather conditions during the second year improved PAP, which was due to the absorption of more light by the leaves and the prolongation of photosynthetic activity [56,57]. Carbohydrate production during grain filling improved, and this was more evident in the second year of the experiment for OF and INT (especially for INT implementing B-W). This improvement may be due to the gradual release of mineral nitrogen from decomposing crop residues and organic fertilisers, as well as the growth and expansion of the photosynthetic green surface area owing to the availability of mineral resources, especially at the flowering stage of the [58,59]. Under OF, we observed a remarkable increase in physiological sinks but only a slight improvement in physiological resources (leaves) in the second year of the experiment. PAP alone cannot supply the requirements of the physiological sinks during the grain filling. The ability of leaves to produce enough photoassimilates is strongly linked to the contents of *Chl-a* and *Chl-b*, which were low under OF [60,61]. Under INT, organic and inorganic inputs were managed in a manner that balanced the plant nutrient requirements in the soil and plant growth stages. This equilibrium in growth stages enabled the plants to use both PAP and DMR to support the physiological sinks [62,63].

OF implementing M-W produced only low amounts of DMR in the second year of the experiment. This may be due to the low production of physiological sources, e.g., leaves before anthesis. Thus, the grains in the M-W crop rotation benefited from PAP during grain filling [64]. The rotation of legumes with wheat under IF, particularly in the second year of the experiment, produced higher grain numbers, which was due to the contribution of the DMR in grain filling. The relative contribution of DMR to grain yield is basically related to source/sink interaction during grain filling [65].

*4.2. Effect of Year and Treatment Interactions on Wheat Yield Components*

Although the content of *Chl-a* and consequently the PAP were higher under IF, the grain yields under IF (B-W) and INT (B-W and S-W) did not differ significantly (slightly higher values under IF as expected) in the second year of the experiment. Therefore, it is possible to decrease the application of chemical inputs by replacing them with organic

fertilisers without drastically reducing grain yield. Applying organic matter together with inorganic fertiliser improves the fertility and bio-chemical properties of soil, which significantly increases wheat grain number and yield [66]. Under INT management, higher levels of *Chl-b* and DMR of the stored pre-anthesis carbohydrates, particularly in the S-W crop rotation system, are the reasons for increased grain yield in the second year. Furthermore, although INT management relies on the availability of inorganic nutrients at the beginning of plant growth, the gradual improvement in soil physical characteristics and the permanent release of nutrients from organic resources (e.g., crop residues, compost, and vermicompost) result in crop yield improvement [67].

Despite the low values of grain number and yield under OF in the first year, these parameters improved in the second year of the experiment. Generally, organic and biological inputs need sufficient time to release crop-available nutrients in the soils to the plant (e.g., due to the conversion of plant residues into mineral components) [68,69]. Therefore, this management strategy requires sufficient time to build up plant nutrients in the soil [70] that ultimately improve crop yield. By comparing the number and weight of grains under OF, we demonstrated a compensatory effect of DMR together with PAP at the grain filling stage, which increased grain yield by improving 1000-grain weight. Considering that the numbers of spikes per square metre and 1000-grain weights did not differ significantly across all the farm management practices (Figure S6a,b), the relatively low yield under OF was mostly due to the low number of grains. This may be due to the slow initial growth of wheat because of low levels of nitrogen from organic compounds [71]. However, the efficacy of both OF and INT was also partially linked to climatic variables, particularly during the grain filling stage (from February to April, Figure 1). For example, wet conditions during the second year accelerated the decomposition rate of crop residues and organic manures, which provided more nutrients for plant growth under OF and INT.

Various summer crops in in wheat-based rotation systems had different effects on yield and its components due to variations in the root and shoot residues of these crops. These may have impacts on improving soil nutrient availability, especially nitrogen. In this study, all summer crops had a positive effect on the grain number and yield of wheat compared to the F-W crop rotation system. Other studies have found similar results, i.e., the yields of wheat in rotation with soybean and maize were higher than that monocropped wheat [23,26]. In contrast, ref. [25] observed no significant difference in grain yield in wheat crops rotated with soybean and maize. The legume–wheat crop rotation system (here, mung bean with wheat) was the most beneficial in terms of grain number, 1000-grain weight, and yield. This was because under this system, nitrogen use was more efficient, due to the symbiosis of legume roots with rhizobium bacteria. Overall, all of these processes associated with organic fertiliser/crop residue management had a positive effect on soil structure and soil quality [36,72,73].

### 4.3. Effect of Year and Treatment Interactions on Wheat Grain Protein Content

All crop rotation systems, except for F-W, did not improve grain protein content under IF compared to those implemented under OF and INT. Nitrogen availability during grain filling is one of the most important factors affecting grain protein content [74]. Under IF management, the high number of grains per spike decreased the contribution of protein allocation to each physiological sink. Furthermore, under IF, inorganic nitrogen fertiliser was applied to the soil at two stages (i.e., planting and tillering) before anthesis, and thus, the crops used a large amount of the absorbed nitrogen to produce the physiological sources and sinks. Therefore, less soil nitrogen was available to the crops at the time of grain filling. Our results contradict those of [75], in which it was reported that the availability of minerals derived from organic fertiliser decomposition was lower during the early stage of wheat growth, which later resulted in reduced grain weight and protein storage. Grain protein content is directly (through nitrogen absorption from the soil) and indirectly (through remobilisation of nitrogen stored in other organs in the pre-anthesis phase to the grain) linked to the availability of sources of nitrogen during grain filling [74]. When organic and

biological compounds (in INT and OF) are used, the decomposition of crop residues into minerals occurs gradually, and therefore, the required minerals (especially nitrogen) are only slowly accessible to plants. This can improve nitrogen availability to crops at the time of grain filling [76].

Our results show that grain protein content was highest when implementing B-W in all three types of farming management. Legume residues have a relatively low C/N ratio (14.7, Table 2), which supports faster rates of decomposition and release of mineral nitrogen into the soil [77]. Thus, legumes can provide nitrogen during grain filling, which improves grain protein concentration compared to other crop rotation systems.

## 5. Conclusions

In this 2-year experiment, all farming management and crop rotation systems produced similar 1000-grain weight values, although this parameter increased from the first experimental year to the next. The contents of *Chl-a*, *Chl-b*, and total Chl, the levels of DMP and PAP, as well as grain number per spike and grain yield were remarkably affected by the type of farming management. Under INT, particularly with B-W and S-W, we observed improved levels of *Chl-b*, resulting in higher PAP as well as the remobilisation of stored pre-anthesis carbohydrates to the grains, leading to high wheat grain yield in the second year of the experiment, despite a 50% reduction in chemical inputs. Moreover, returned crop residues and organic fertiliser in the soil under INT resulted in the highest grain protein content among the treatments. Among the three wheat crop rotations with sesame, mung bean, and maize, the B-W rotation had the most positive effect on wheat yield, followed by S-W, M-W, and F-W. For this region, this is a novel and promising finding, because it contradicts the assumption of farmers that sesame is an exhaustive crop that negatively affects soil properties and hence wheat yields due to its aggressive tendency to consume soil nutrients at different depths. This study, however, suggests that future investigations should examine different varieties of wheat in rotation with summer crops under integrated farming management over larger spatial scales. Such investigations should be conducted as long-term experiments, with the aim of finding the most compatible wheat variety for INT management.

**Supplementary Materials:** The following supporting information can be downloaded at: https://www.mdpi.com/article/10.3390/agronomy13041007/s1.

**Author Contributions:** Data curation, A.G. and T.G.; conceptualization, A.G.; methodology, A.G., A.A. and A.R.; data processing, A.G. and T.G.; writing—original draft, A.G.; writing—review and editing, B.K. and T.G.; data collecting, A.G. and A.A. All authors listed contributed to the article and approved it for submission. All authors have read and agreed to the published version of the manuscript.

**Funding:** This research was funded by Shahid Chamran University of Ahvaz, grant number SCU.AA1401.96.

**Institutional Review Board Statement:** Not applicable.

**Informed Consent Statement:** Not applicable.

**Data Availability Statement:** Data will be accessible and shared via email or link address. To obtain the data of this research, send your request by sending an email to gorooei@uni-bonn.de.

**Conflicts of Interest:** The authors declare no conflict of interest.

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
