# Peer review of "The Effect of Farming Management and Crop Rotation Systems on Chlorophyll Content, Dry Matter Translocation, and Grain Quantity and Quality of Wheat (Triticum aestivum L.) Grown in a Semi-Arid Region of Iran"

_agronomy, doi:10.3390/agronomy13041007_

Round 1

Reviewer 1 Report

M & M:

Line 135-136, the experiment includes two factors, i.e. main factor of farming managements and subfactor of crop rotation systems (12 treatments)

Table 1, what's kind of compost used in the present experiment? Crop residues return to soil, please change the unit to t ha-1. Pleas listed the total rates of N, P, and K including organic and inorganic fertilizer in the Table 1 or in the supplementary materials.

In Equation 6, what's the meaning of 5.7?

Results:

Figure 2 it is suggested to split Fig.2a into two figures, each figure represents one year data. the same as b and c and other Figures if necessary. Moreover, which letters in the figure represented difference at P<0.05 and P<0.01, respectively?

Figure 3, which year data was presented in Figure 3? The same as other figures.

Minor corrections:

Line 54-55, appropriate should not be italic.

Line 70 delete (2019)

Line 72, should be wheat-summer crops?

Author Response

Response to comments for article reference: ID- Agronomy-2301590

Title:

The effect of farming management and crop rotation systems on chlorophyll content, dry matter translocation, and grain quantity and quality of wheat (Triticum aestivum L.) grown in a semi-arid region of Iran

Answer to the comments of referee #1

We are thankful to the reviewer for the comments and suggestions which helped us to improve the quality of the manuscript. We have revised the paper accordingly. Here we summarized the changes that we have made in the revised version of the manuscript.

Line 135-136, the experiment includes two factors, i.e. main factor of farming managements and subfactor of crop rotation systems (12 treatments).

We rewrote the sentences based on the referee’s comment. The new lines 189-190 now read as:

 “The experiment includes two factors, i.e. main factor of farming managements and subfactor of crop rotation systems (12 treatments).”

Table 1, what's kind of compost used in the present experiment? Crop residues return to soil, please change the unit to t ha-1. Pleas listed the total rates of N, P, and K including organic and inorganic fertilizer in the Table 1 or in the supplementary materials.

  • The used compost was manure compost. It corrected in the new version of the revised paper.
  • The unit of crop residues changed from g m-2 to t ha-1 (Table 1).
  • The percentage of N-P-K added to the supplementary materials (Table 2).

In Equation 6, what's the meaning of 5.7?

The value of 5.7 refers to the protein factor for wheat. We described it in the revised article and also added the reference we used for that. Between line 208 to 209.

Results:

Figure 2 it is suggested to split Fig.2a into two figures, each figure represents one year data. The same as b and c and other Figures if necessary. Moreover, which letters in the figure represented difference at P<0.05 and P<0.01, respectively?

  • All designed figures divided into two figures based on the year in the entire article.
  • Since the SAS analysis was performed based on Duncan's multiple test at a significance level of P<0.05, we removed P<0.01 for the caption of the figures and kept P<0.05. In addition, the effect of the experimental treatment at both significance levels are given in Tables 2 and 3.

Figure 3, which year data was presented in Figure 3? The same as other figures.

Figures 3, 5 and 7 indicated the interaction effect of farming management and crop rotation system. These figures do not refer to the effect of year. The interaction effect of year with other treatments, if it was statistically significant, is shown in Figures 2, 4 and 6.

Minor corrections:

Line 54-55, appropriate should not be italic.

"Appropriate" was rewritten in non-italic in the revised version of the article. Line 55.

Line 70 delete (2019)

(2019) was removed and all references were checked accordingly. Line 70.

Line 72, should be wheat-summer crops?

The sentence refers to the winter wheat in rotation with summer crops. The sentence was modified in the new version of the article and read now as:

“Similarly, double cropping systems based on winter wheat and summer crops e.g., sorghum (Sorghum bicolor L.) and sesame (Sesamum indicum L.) improved annual net production and long-term sustainability compared to a summer-fallow rotation system.”

Reviewer 2 Report

Dear Authors,

Congratulations for the good work.

Please check one more time the style of the references.

Best regards

Author Response

Answer to the comments of referee #2

We really thank the reviewers for their positive comment and for putting our work into context.

Please check one more time the style of the references.

All the references in the text of the article were checked and their style was also checked.

Reviewer 3 Report

The paper has novelty and contributes good information for suitable farming management approaches in the semi-arid area. The paper is well written and organized. I suggest to seperate the Discussion into several parts with subheadings, then it will be more readable.

Author Response

Answer to the comments of referee #3

The paper has novelty and contributes good information for suitable farming management approaches in the semi-arid area. The paper is well written and organized.

We really thank the reviewers for their positive comment and for putting our work into context.

I suggest to separate the Discussion into several parts with subheadings, then it will be more readable.

The discussion section was divided into 3 parts. Lines 360-361, 407, and 450.

4.1. Effect of year and treatment interactions on chlorophyll contents and dry matter translocation

4.2. Effect of year and treatment interactions on wheat yield components

4.3. Effect of year and treatment interactions on wheat grain protein content
